# Higher Leg and Trunk Muscle Activation during Balance Control in Copers versus People with Chronic Ankle Instability and Healthy Female Athletes

**DOI:** 10.3390/sports10080111

**Published:** 2022-07-22

**Authors:** Mina Karbalaeimahdi, Mohammad Hossein Alizadeh, Hooman Minoonejad, David G. Behm, Shahab Alizadeh

**Affiliations:** 1Department of Health and Sport Medicine, Faculty of Sport Sciences and Physical Education, University of Tehran, Tehran 1417935840, Iran; mina.karbalaii@ut.ac.ir (M.K.); alizadehm@ut.ac.ir (M.H.A.); h.minoonejad@ut.ac.ir (H.M.); 2School of Human Kinetics and Recreation, Memorial University of Newfoundland, St. John’s, NL A1C 5S7, Canada

**Keywords:** electromyography, neuromuscular control, ankle sprain, coper, Cumberland ankle instability tool

## Abstract

**Highlights:**

**Abstract:**

More than 70% of people with ankle sprain experience chronic ankle instability. However, some people are well adapted to this damage (copers) and do not suffer from chronic ankle instability (CAI). This cross-sectional study involved 34 female athletes, who were classified into three groups (athletes with CAI, copers, and healthy athletes) and tested on a Biodex Balance System. Surface electromyography (EMG) and balance scores were monitored. The coper and healthy group exhibited higher medial gastrocnemius (MG) EMG activity during unstable balance conditions. The rectus abdominus (RA) in the coper group and rectus femoris (RF) in the healthy group showed greater EMG activity compared to CAI during unstable conditions. During stable conditions, the coper group showed greater RA EMG activity compared to CAI, as well as higher tibialis anterior (TA) EMG activity compared to the healthy group. Additionally, balance error scores were higher in the CAI group than those in the healthy group under unstable conditions. In conclusion, decreased EMG activity of the MG, RF, and RA in CAI athletes may contribute to impaired balance in these individuals. The increased EMG activity of the MG, TA, and RA in copers might result in more trunk and ankle stability.

## 1. Introduction

Ankle sprains are one of the most prevalent musculoskeletal injuries, with athletes in the United States reporting ankle sprain accounting for 15% of total injuries [1]. More than 70% of individuals that experience initial lateral ankle sprain (LAS) develop chronic ankle instability (CAI) [2] that leads to compromised joint capsule, ligament, tendon, and muscle integrity. LAS will damage the mechanoreceptors causing disruption to the proprioception system [3], leading to faulty postural [4,5] and coordination control [6,7]. However, certain individuals, even with prior LAS, who are referred to as copers (i.e., the ability to cope with or accommodate LAS), do not experience symptoms of CAI or recurrent injury [8,9]. Copers are defined as individuals who have not suffered from a recurrent ankle sprain for 12 months after their initial LAS incident [9]. However, the mechanisms involved which enable copers to regain functionality similar to the uninjured individuals are still unclear [10].

Compared to healthy individuals, patients with a history of ankle sprain during single-leg stance demonstrate a dominant postural control strategy from the hip rather than the ankle [11]. Research has shown that healthy individuals are able to maintain postural sway stability by predominately relying on ankle muscle receptors [12]. Therefore, the ankle’s proprioceptors can help people maintain their balance during an unperturbed stance. It is known that during an unperturbed stance, the body’s sway is correlated with the amount of ankle rotation, which explains why muscles surrounding the ankle are able to provide sensory information necessary to maintain balance [13]. It is suggested that proprioceptive receptors within the ankle may be damaged as a result of an initial sprain, and thus can lead to changes in muscle spindle activation and sensitivity [7]. The ankle strategy is linked to the fine tuning of static postural control, whereas the hip strategy is used to counterbalance more substantial postural control disturbances [14].

In the case of copers, they showed a comparable postural control strategy to healthy individuals during open and closed eyes in a single-leg stance test [9,11,15]; however, information regarding muscle activation during dynamic balance tasks in these individuals is limited. The CAI group has shown altered activity of the muscles supporting the ankle due to nerve and muscle damage. Recent studies reported that CAI individuals showed significantly less peroneus longus activity and shorter peroneus longus latency compared to the coper group in dynamic tasks [16,17]. In a study by Kwon et al., it was reported that muscle activity in the CAI group is different compared to the healthy and coper groups during single-leg balance with eyes closed [18]. Muscle strategies used during dynamic balance tasks are phase dependent between CAI, copers, and healthy individuals. Thus, understanding how muscles are recruited during dynamic controlled tasks can add to our understanding of how copers differ from CAI in muscle recruitment. Individuals with CAI usually show changes in the activation of the muscles around the ankle and sometimes the proximal muscles, and also present changes in functional tasks, including balance [19,20].

Previous research has shown that during various balance tasks, people with CAI exhibited atypical levels of muscle activation than copers [3,7,8,11,21,22,23,24]. Compared to people with CAI, copers demonstrate greater tibialis anterior (TA) electromyography (EMG) activity, suggesting greater ankle stability [21]. In another study, the difference in muscular activation was shown in the gluteus maximus muscle, with the CAI patients exhibiting lower EMG activation compared to healthy people and copers during a balance test [19]. The difference in EMG activity between these groups can be related to the different postural control strategies that these groups incorporated during a balance task [4]. 

Thus, it was hypothesized that the muscles involved in maintaining balance are different between individuals with CAI, compared to the copers group, and this can lead to balance disorders in these individuals. Secondly, it was also hypothesized that muscle activity of these individuals can differ between stable and unstable surfaces while balancing on them. Another hypothesis contends that individuals with CAI have reduced ankle muscle activity due to injury, and are more likely to activate the proximal muscles to compensate for this deficiency. By examining these cases, we can have a better understanding of copers for adapting to injury, which will be important in improving balance control and problems in people with CAI. Therefore, the purpose of this study was to investigate the balance and neuromuscular activity of the muscles involved in postural control strategies, such as tibialis anterior (TA), medial gastrocnemius (MG), rectus femoris (RF), biceps femoris (BF), erector spinae (ES), and rectus abdominis (RA), during a single-leg stance between CAI, copers, and healthy athletes.

## 2. Materials and Methods

### 2.1. Participants

A sample of thirty-four female athletes were allocated to three groups: CAI, copers, and healthy athletes (Table 1). The sample size was calculated using G-power software with alpha levels set to 0.05, with a moderate effect size of 0.33 and power of 0.9. The inclusion criteria for the participants were as follows: (1) age range from 18 to 30 years; (2) a minimum score of 24 or higher on the Cumberland ankle instability tool (CAIT) was considered for athletes to be placed in the coper group, and a score of 24 or less for athletes to be classified in the CAI group [25]; (3) no history of LAS for healthy athletes; (4) no new or recurrent ankle sprain during the previous year with no disability and episodes of instability since the time of LAS for copers; and (5) a history of at least two instability episodes in the last six months for the CAI group [8]. Participants were excluded from the study based on the following criteria: (1) any history of lower limb surgery; (2) athletes suffering from vestibular system disorders (balance problems); (3) any alcohol and/or drug consumption within the 24 h prior to the study. The study was approved by the local ethical committee of Tehran University (IR.SSRI.REC.1396.194) and written signed consent was obtained from the participants.

### 2.2. Experimental Procedure

Five minutes of self-paced walking was completed as a warmup. All participants wore sports clothes and shoes during warm-up, and short tops and shorts during measurements. After the warmup, participants were familiarized with the balance tests at two levels of difficulty (level 3: unstable and level 12: stable) on the Biodex Balance System (BBS) system. All participants were given a practice session using the same testing protocol, as a familiarization with the balance platform. For the CAI and coper groups, all measurements were taken from the injured limb, and for healthy athletes (control group) from the dominant leg, which was defined as the leg used to kick a soccer ball [26]. Athletes were instructed to remain barefoot on the center of the platform, with the opposite leg bent from the knee at a 90° angle, avoiding contact with the testing leg or platform, with their hands placed on their chest (Figure 1). The foot was positioned in the centre of the platform, following the guidelines specified by BBS. Foot position was recorded and replicated for each person for each trial. BBS measures the degree of tilt about each axis during dynamic action in closed-chain conditions, provides stability indices, and calculates the anterior–posterior index (API), the medial–lateral stability index (MLI), and the overall balance index (OBI). The lower these values, the better the individual’s balance is. After the familiarization, the participants were asked to complete the BBS test at the two difficulty levels of stability in a randomized order, with EMG recording from the TA, MG, RF, BF, ES, and RA. Each level consisted of three 20-s trials with a 10-s rest between each trial (Figure 2). This study was carried out at the Laboratory of Health and Sports Medicine Department, University of Tehran, Tehran, Iran.

### 2.3. Postural Stability

Balance scores was recorded at two levels of stability (level 3: unstable and level 12: stable) using the BBS athlete single-leg test. If the postural stability was not met during the test, the results from the related trial were nullified and the participant repeated the trial once more. The average scores of the anterior–posterior and medial–lateral indices from three trials were used for statistical analysis. The BBS measures the amount of body tilt around each axis in dynamic and closed-chain conditions, provides stability indices, and calculates the stability indices of anterior–posterior, medial–lateral stability, and the overall stability index (it measures the platform displacement variance in all directions). Lower values indicate better balance of the individual.

**Figure 2 sports-10-00111-f002:**
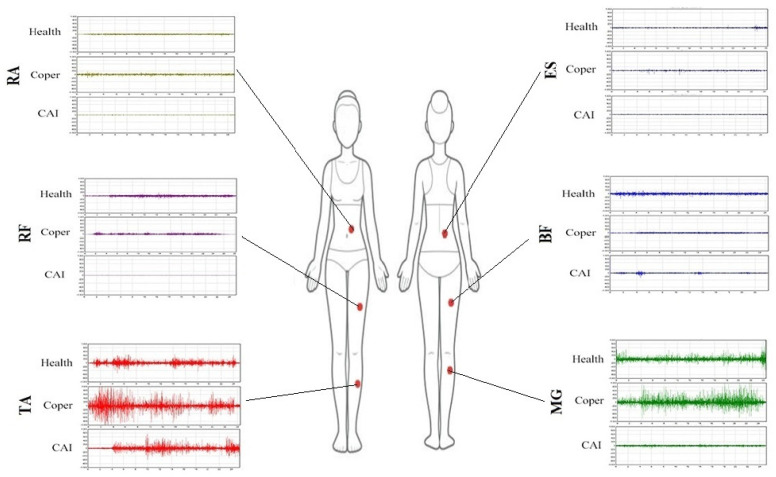
An example of the raw wave of muscle activities (*n* = 1).

### 2.4. Electromyography (EMG) Measurement

EMG (ME6000 16-channel EMG, Megawin, Finland) activity from TA, MG, RF, BF, ES, and RA were recorded at a sampling frequency of 1000 Hz (bandwith = 8–500 Hz, common mode rejection ratio (CMRR) = 110 dB, input impedance = 10 Gohm, Gain: 305). The skin was prepared by removing excess hair, abrading, and cleaning with an alcohol swab. The electrodes (self-adhesive Ag/AgCl bipolar, SKINTACT ECG Electrodes, Austria) were placed on the TA, MG, RF, BF, ES, and RA muscles following SENIAM guidelines, as illustrated in Figure 2 [27,28,29]. All EMG data were band-pass filtered (10 to 350 Hz) and notch filtered (60 Hz at 1-Hz width) using a Butterworth filter (4th order, zero-phase lag) in the Megawin software package used for the EMG processing. The data were rectified and smoothed by taking the root mean square average of the EMG signal and using a 50-millisecond sliding window function. EMG values were normalized to their respective peak root-mean-square value during an MVC (EMG value/MVC %). Finally, the MVCs from the TA, MG, RF, BF, ES, and RA muscles were obtained using the method described previously by Konrad [28] on the same day.

### 2.5. Statistical Analysis

The Brown–Forsythe F-test was used to compare the difference in height, weight and body mass index (BMI) between the three groups due to unequal sample sizes between groups. Non-parametric Kruskal–Wallis H tests were used for EMG activity for each muscle and BBS scores at both level 3 and level 12 of BBS. If a significant effect was found, a post hoc Mann–Whitney U test was carried out to identify the differences among groups (CAI, copers, and healthy athletes). Furthermore, Wilcoxon signed ranks tests were used to compare EMG activity for each muscle and BBS scores between two difficulty levels of the BBS. Alpha level was set to 0.05 for Kruskal–Wallis H tests. A Bonferroni correction on the alpha level was calculated for Mann–Whitney U and Wilcoxon signed ranks tests to reduce type I error. Effect sizes were calculated for the non-parametric tests (r) in which the values were considered small (0.1–0.29), medium (0.3–0.49), and large (>0.5) [30]. All statistical analyses were conducted using SPSS version 21.

## 3. Results

### 3.1. EMG Activity during Unstable Condition

MG activity was higher in the coper (↑133%) and healthy athletes (↑116%) groups compared to the CAI group. RA activity in the coper group was elevated compared to the CAI group (↑77%). RF activity was higher in the healthy athletes (↑148%) compared to the CAI group (Table 2). No significant difference was found between groups for TA, BF, and ES muscle activity during unstable conditions (*p* > 0.05).

### 3.2. EMG Activity during Stable Condition

The coper group showed higher (↑68%) RA EMG activity compared to the CAI group. Additionally, TA muscle had greater EMG activity in the coper group (↑105%) compared to the healthy athletes. RF EMG activity was greater in healthy athletes (↑166%) compared to the CAI group (Table 2).

### 3.3. EMG Activity between Stable and Unstable Conditions

EMG muscle activity of the TA in the CAI (↑76%) and healthy (↑64%) groups; BF in the CAI (↑39%) group; RF in the CAI (↑46%) and coper (↑47%) groups; ES in the CAI (↑54%) and healthy (↑11%) groups; and RA in the CAI (↑6%) and coper (↑12%) groups had significantly higher EMG activity from level 12 (stable) compared to level 3 (unstable) BBS (Table 3). The MG did not show a significant difference in EMG activity between level 3 and level 12 on the BBS. No additional differences were noted for other muscles during the stable condition (*p* > 0.05).

### 3.4. Balance Error Scores during Unstable Condition

The CAI group showed increased (H_(2)_ = 7.346, *p* = 0.025) medial–lateral balance error scores (↑142%) compared to the healthy group (U = 27.5, *p* = 0.011, r = 0.52). The overall balance error score was greater (H_(2)_ = 8.051, *p* = 0.018) in CAI (↑106%) compared to the healthy athletes (U = 26, *p* = 0.008, r = 0.52; Table 4). No further differences were observed between the CAI, coper, and healthy groups (*p* > 0.05).

### 3.5. Balance Error Scores during Stable Conditions

The copers achieved overall higher balance error scores (H_(2)_ = 7.341, *p* = 0.025) compared (↑44%) to the healthy athletes (U = 21, *p* = 0.015, r = 0.52). No further difference was observed between the CAI, coper, and healthy groups (*p* > 0.05).

### 3.6. Balance Error Scores between Stable and Unstable Conditions

Higher values were evident for medial–lateral balance error scores for copers (↑143%), CAI (↑124), and healthy (↑60%) groups in the unstable condition. Greater values were found for anterior–posterior scores for copers (↑42%) and CAI (↑49%), as well as overall balance for copers (↑102%) and CAI (↑82%) in the unstable condition (Table 4). 

## 4. Discussion

The primary findings of this study were that individuals with CAI showed a large magnitude (effect size) decrease in EMG activity for proximal muscle groups (MG) and distal (RF and RA) to the ankle during a single-leg stance. A secondary finding was that both medial–lateral and overall balance during unstable conditions demonstrated significant, large magnitude, higher (decreased balance) indices in the CAI group than the healthy controls.

Previous studies showed a decrease in ankle, knee, and hip EMG activity in people with CAI during functional balance tasks (single-leg balance with closed eyes, star test, and lateral hop) [7,22]. Similar to our results, one study showed lower MG EMG activity in individuals with CAI during single-leg stance when compared with healthy controls [31]. In contrast, one study reported greater MG EMG activity in the CAI group compared to the copers and healthy group [18]. In their study, Kwon et al. reported that, during a single-leg balance test with eyes closed, the CAI group had higher fibularis longus and MG activity than the healthy and coper groups [18]. Difference in testing protocols can be designated as the source of this discrepancy, with the Kwon study conducting balance tests in participants with their eyes closed. The lack of visual input has been known to disrupt balance [18], thus requiring more muscular contribution to achieve stability compared to having visual feedback [32], as individuals would rely more on proprioception feedback mechanism for balance which leads to altered muscle activation. However, difference in muscular EMG activation is not limited to the ankle, and can also reach the upper segments.

In addition to the muscles supporting the ankle joint, muscles distal from the ankle (i.e., RF and RA) showed decreased EMG activity in people with CAI. A previous study partially supports these findings, reporting that patients with CAI demonstrated a decreased hip and ankle muscle activity during performance of the Star Excursion Balance test (SEBT) [25]. The alterations in the sensory information after an ankle sprain incident has been suggested to cause central neural adaptation, in which the EMG activity of the proximal and distal muscles would change [33,34,35,36,37]. Additionally, an increase in balance task difficulty has been shown to recruit muscles from the upper kinetic chain such as the hip region, as well as the muscles surrounding the ankle joint [38]. Based on these findings, it is expected that the amount of muscle activation in the proximal, as well as the distal, kinetic chain would differ between individuals who suffer from CAI and those who do not, due to the sensory information alteration when attempting to balance on an unstable surface. 

Another finding was that no significant difference in balance indices or EMG activity in unstable conditions were found between healthy controls and copers. Very little work has been done on the BBS; however, other results support the findings of this study [19]. Regarding changes in surface stability, the unstable surfaces, compared to stable surfaces, were found to induce greater muscle activation in all groups. Where copers demonstrated a higher muscle activation in the hip region (RA and RF), the healthy group incorporated greater muscle activation from the ankle and trunk region (TA and ES), while balancing on unstable surfaces compared to stable surfaces. The CAI group showed higher muscular activation from the ankle, hip, and trunk region (TA, BF, RF, ES, and RA) as a result of increased balance difficulty. These muscular patterns suggest that copers employ a greater hip strategy as the difficulty of balance level increases, whereas healthy athletes adopt more ankle and trunk strategies [39]. Based on this idea, it seems that the CAI group incorporates all of the balance strategies (i.e., ankle, hip, and trunk) as the difficulty of balance level increases. 

With regards to balance performance, two studies found no significant difference in the performance scores during a SEBT between copers and healthy individuals [40,41]. For healthy and CAI individuals, there was no difference in balance error scores between the CAI and healthy athletes, which is similar to findings from a previous study [42]. Similarity of strategies adopted between copers and healthy individuals can also be seen during other challenging tasks, such as landing and cutting movements [43]. It is speculated that the copers develop compensatory motor control mechanisms [9] that would allow them to perform similarly to unaffected individuals, but the details of this matter are not yet fully understood. One limitation of this study was the lack of use of both sexes. Inclusion of male coper participants and comparing them to a female coper population may better suggest how neuromuscular activity among different lower extremity muscles compares between sexes.

## 5. Conclusions

The results of this study demonstrated that the EMG activity of the CAI group was lower than the other two groups during single-leg stance. Individuals with CAI presented a decrease in EMG activity of muscles both distal (RF and RA) and proximal (MG) to the ankle during single-leg stance in the unstable condition; however, it is difficult to conjecture with certainty which strategy in these individuals is more likely to contribute to maintaining the balance with a single-leg stance. Moreover, people with CAI had balance deficits in the medial–lateral direction. Reduced muscle activity in individuals with CAI could be one of the reasons for the balance deficits in these individuals. Increased EMG activity of the MG, TA, and RA in copers might contribute to compensatory mechanisms, which results in greater trunk and ankle stability.

## Figures and Tables

**Figure 1 sports-10-00111-f001:**
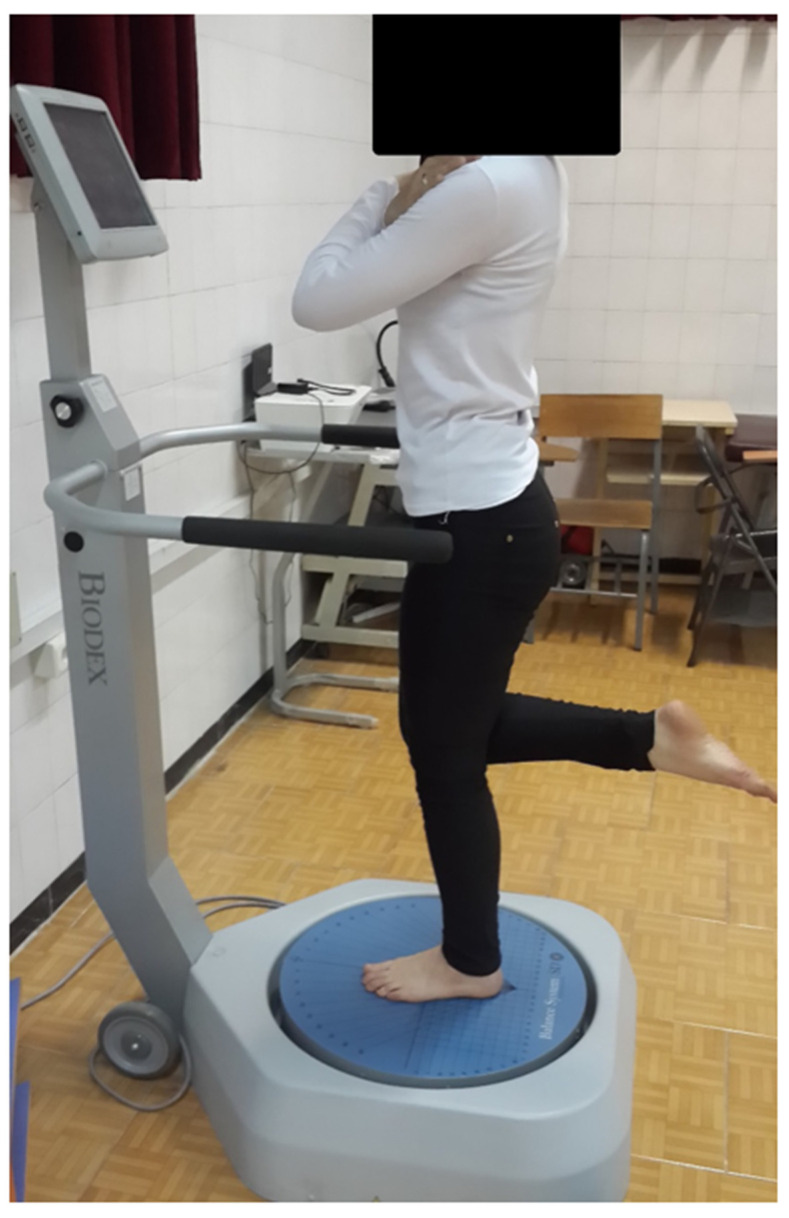
Subject’s position on the balance platform.

**Table 1 sports-10-00111-t001:** Mean ± SD of anthropometric and demographic data for copers (*n* = 10), chronic ankle instability (CAI, *n* = 13), and healthy athletes (*n* = 11).

	Copers	CAI	Healthy	Sig.
Age (years)	21.9 ± 1.37	22.3 ± 4.21	24.5 ± 4.1	0.174
Height (cm)	165.9 ± 4.7	166.7 ± 5.8	165.7 ± 5.3	0.873
Body mass (kg)	56.2 ± 5.7	59 ± 7.4	57 ± 6.9	0.572
BMI (kg.m^−2^)	20.5 ± 2.7	21.2 ± 2.6	20.7 ± 2.5	0.784

**Table 2 sports-10-00111-t002:** Mean ± SD EMG activity (%MVC) of different muscles at two levels of balance difficulty (3 and 12) on the Biodex Balance System (BBS) for copers, chronic ankle instability (CAI) patients, and healthy athletes.

	Coper	CAI	Healthy	Sig.	Effect Size (r)
Medial gastrocnemius (level 3)	35 ± 22	15 ± 4	33 ± 19	0.015 * 0.005 ^†^	0.5 ** 0.53 ^††^
Rectus abdominis (level 3)	9 ± 5	5 ± 3	6 ± 3	0.013 *	0.51 **
Rectus femoris (level 3)	10 ± 5	6 ± 4.2	14 ± 5	0.008 ^†^	0.53 ^††^
Rectus abdominis (level 12)	8 ± 5	5 ± 3	6 ± 3	0.015 *	0.5 **
Tibialis anterior (level 12)	18 ± 13	9 ± 4.2	9 ± 5	0.013 ^‡^	0.53 ^‡‡^
Rectus femoris (level 12)	8 ± 5	4 ± 4	11 ± 5	0.001 ^†^	0.67 ^††^

* significant difference between coper and CAI (*p* < 0.05), ^†^ significant difference between healthy and CAI (*p* < 0.05), ^‡^ significant difference between coper and healthy (*p* < 0.05), ** r effect size calculated between coper and CAI, ^††^ r effect size calculated between healthy and CAI, ^‡‡^ r effect size calculated between coper and healthy.

**Table 3 sports-10-00111-t003:** Mean ± SD EMG activity (%MVC) of different muscles between two levels of balance difficulty (3 and 12) on the Biodex Balance System (BBS) for copers, chronic ankle instability (CAI) patients, and healthy athletes.

	BBS Level	Coper	CAI	Healthy	Sig.	Effect Size (r)
Tibialis anterior	3	16 ± 13	16 ± 7	14 ± 9	0.003 * 0.004 ^†^	0.86 ** 0.85 ^††^
	12	18 ± 13	9 ± 4	9 ± 5		
Biceps femoris	3	4 ± 3	3 ± 2	3 ± 2	0.015 *	0.67 **
	12	4 ± 3	2 ± 2	3 ± 2		
Rectus femoris	3	10 ± 5	6 ± 4	14 ± 5	0.006 * 0.017 ^‡^	0.76 ** 0.75 ^‡‡^
	12	8 ± 5	4 ± 4	11 ± 5		
Erector spinae	3	6 ± 4	5 ± 3	5 ± 3	0.011 * 0.041 ^†^	0.66 ** 0.61 ^††^
	12	5 ± 6	3 ± 1	5 ± 3		
Rectus abdominis	3	9 ± 5	5 ± 3	6 ± 3	0.018 * 0.025 ^‡^	0.66 ** 0.71 ^‡‡^
	12	8 ± 5	5 ± 3	6 ± 3		
Medial gastrocnemius	3	35.35 ± 22.1	15.97 ± 4.1	32.9 ± 18.56	NS	NS
	12	31.56 ± 20.43	20.68 ± 13.13	32.56 ± 22.9		

* significant difference for CAI between two levels (3 vs. 12), ^†^ significant difference for healthy between two levels (3 vs. 12), ^‡^ significant difference for copers between two levels (3 vs. 12), ** r effect size for CAI group between two levels (3 vs. 12), ^††^ r effect size for healthy group between two levels (3 vs. 12), ^‡‡^ r effect size for copers between two levels (3 vs. 12), NS: non-significant.

**Table 4 sports-10-00111-t004:** Mean ± SD Biodex Balance System (BBS) error scores between two levels of balance difficulty for copers, chronic ankle instability (CAI) patients, and healthy athletes.

	BBS Difficulty Level	Coper	CAI	Healthy	Sig.	Effect Size (r)
Medial–lateral	3	2.3 ± 1.6	2.1 ± 1.4	0.8 ± 0.4	0.002 * 0.026 ^†^ 0.008 ^‡^	0.84 ** 0.67 ^††^ 0.83 ^‡‡^
	12	0.9 ± 0.4	0.9 ± 0.5	0.5 ± 0.2		
Anterior–posterior	3	1.3 ± 0.7	1.8 ± 0.7	1.1 ± 0.9	0.013 * 0.021 ^‡^	0.64 ** 0.73 ^‡‡^
	12	0.9 ± 0.3	1.2 ± 0.4	0.8 ± 06		
Overall	3	3 ± 1.3	2.9 ± 1.4	1.4 ± 0.9	0.003 * 0.008 ^‡^	0.82 ** 0.73 ^‡‡^
	12	1.4 ± 0.2	1.6 ± 0.4	1.1 ± 0.5		

* significant difference for CAI between two levels (3 vs. 12), ^†^ significant difference for healthy between two levels (3 vs. 12), ^‡^ significant difference for copers between two levels (3 vs. 12), ** r effect size for CAI group between two levels (3 vs. 12), ^††^ r effect size for healthy group between two levels (3 vs. 12), ^‡‡^ r effect size for copers between two levels (3 vs. 12).

## Data Availability

Not applicable.

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
