# Peer review of "Higher Leg and Trunk Muscle Activation during Balance Control in Copers versus People with Chronic Ankle Instability and Healthy Female Athletes"

_sports, 2022, doi:10.3390/sports10080111_

Round 1

Reviewer 1 Report

Title: 

The title does not accurately reflect the content of the article since a comparison is also made with healthy persons with no medical history indicated to (LAS), and the type of study to which it refers could also be indicated. The population used in this study is only female, so we cannot refer to people in general, but should refer to the sex of the study, female in this case.

Abstract:  

- Electromyography, they make use of acronyms without putting the word it refers to.

- It is said that the Cumberland ankle instability tool was used as part of the method, but it was only used for group discrimination.

- In form it is well written, it deals with all the points of the work.

 Introduction

- At the beginning of the introduction reference is made to an injury rate for 2010 and only in the United States, but this is not indicated in the text and gives the impression that it is a global rate.

- Reference 7 is not well arranged in the text, it is not properly arranged and it is not clear what it refers to.

- It talks about postural control in unstable situations rather than in unstable situations.

- It is placed before talking about specific muscular activations of both tibialis anterior and gluteus maximus, to a generalization regarding the neuromuscular receptors of the ankle, which would explain the changes in the muscular activation previously described.

- The comprehension of the last paragraph is difficult due to the absence of semicolons.

-Some more references are needed to support what is said "Individuals with CAI usually show changes in the activation of the muscles..............." and to support the 3 hypotheses that are described prior to the hypothesis that is the object of the study.

- Regarding the structure of the introduction, it should be improved to speak from the most general to the most particular, the first paragraph is in the correct situation of the introduction, but the second, in part should be almost at the end of the introduction before launching the hypothesis and study question.

2. Materials and Methods

2.1 Participants

-The inclusion criteria are well founded, with the scientific literature.

- In the exclusion criteria, criterion number 3 is missing, it goes from criterion 2 to criterion 4.

-Exclusion criterion number 4, is it checked in any way?

-The specification of in which leg the measurement of each group is carried out should not be in this section of the research, but in the following one.

2.2 Experimental procedure

- It would be good to indicate the place where the study was carried out, laboratory, it is supposed, but it is not specified.

- The clothing and footwear used by the participants, both in the warm-up and in the performance of the tests, are not specified.

- It is not specified how is the familiarization of the participants with the BBS test, which may give more time to adapt to the test between participants.

- The part of the text that refers to the MVC of the EMG should be described in section 2.4 Electromyography (EMG) measurement. 

-Figure 1 as above should be located in section 2.4 Electromyography (EMG) measurement.

2.3 Postural stability

- The specification of the footwear should be in the previous section 2.2 Experimental procedure.

- What happens if the postural stability criteria are not met? Is the test repeated or is the test null? Not specified.

- Does the angle of foot placement on the platform not matter?

2.4 Electromyography (EMG) measurement

- This section is well explained.

- For a better reproduction of the study and better assessment, the exact location for each of the muscles could be known and not only for the electrodes used, but also for the electrodes used in the study. 

- The SENIAM guide is referenced, but only to refer to the electrodes used.

- Add the section on calculation of the CVM and find out whether the CVM and the test were performed on the same day, taking advantage of the same location of the CVM for the subsequent performance of the test.

2.5 Statistical analysis

-This section is correctly described

3. Results

3.1 EMG activity during unstable condition

- Unstable situations are mentioned in the title of the section, but data are shown in Table 2 for both unstable situations (Level 3) and stable situations (level 12).

- Only 3 muscles are mentioned, Medial gastrocnemius (MG), Rectus abdominis (RA) and Rectus femoris (RF); the remaining muscles are omitted and it is not specified whether or not there are differences in these muscles between groups.

- It does not follow a line in the presentation of the data, the MG compares it between coper and healthy with CAI, the RA compares coper versus CAI and RF compares healthy with CAI.

3.2 EMG activity during stable condition

- The part of table 2 referring to stable situations (Level12) should be indicated in this section.

- The font format used in this section is not as it should be, it is in italic format as if it were an image caption.

- Only 3 muscles are mentioned, Tibialis anterior (TA), Rectus abdominis (RA) and Rectus femoris (RF); the remaining muscles are omitted and it is not specified whether or not there are differences in these muscles between groups.

- does not follow a line in the presentation of the data, the TA compares it between coper and healthy with CAI, the RA compares coper versus CAI and RF compares healthy with CAI.

3.3 EMG activity between stable and unstable conditions

- The values of MG are missing in table 3, since the values of stable conditions (Level 12) are not found in any of the tables or results.

- no differences are specified in healthy subjects for rectus femoris (RF) no significant differences are found.

- no differences are specified for erector spinae (ES) for coper subjects.

3.4 Balance error scores during unstable condition 

- Only comparison data between CAI and healthy subjects are presented; some reference to copers is missing.

- A table of results can be added, to facilitate the understanding of the data.

3.5 Balance error scores during stable conditions

- Only comparison data between copers and healthy subjects are presented, some reference to CAI is missing.

- A results table can be added for these data to improve their visualization.

3.6 Balance error scores between stable and unstable conditions

-This point is well defined.

4. Discussion

- It talks about the decrease in EMG for CAI in single leg stance, without specifying whether in stable or unstable conditions or both.

-It refers to EMG of joints (ankle, knee, hip) rather than muscle groups of those body areas.

- In the fourth paragraph reference is made to other results that support the findings of this study, but no references are specified to support that statement.

5. Conclusions

- This point of the paper is well written except for specifying what type of single leg support it refers to, whether it is stable or unstable or both, besides the medial gastrocnemius no data is shown regarding single leg support in a stable situation in any part of the paper.

6. Bibliography

This section is correctly expressed.

General analysis 

Congratulations to the authors for the study, the subject is very interesting and affects a large population. 

It is suggested that a revision of the material and methods be made, being more exhaustive and concrete in the explanations of the use of electromyography and better detailing the procedures for performing the tests.

In addition, in the results section it is proposed to improve the presentation of the data, not leaving comparisons between groups without making them and following a better line to improve the understanding of the data and make the article easier to understand.

It would also be good to introduce a section on limitations and lines of future research or to add it at the end of the conclusion. 

Author Response

Reviewer #1

Comments and Suggestions for Authors

Title: 

The title does not accurately reflect the content of the article since a comparison is also made with healthy persons with no medical history indicated to (LAS), and the type of study to which it refers could also be indicated. The population used in this study is only female, so we cannot refer to people in general, but should refer to the sex of the study, female in this case.

Response: We would like to thank the reviewer for their valuable comment and suggestion which helps improve the manuscript. We have revised the title of the manuscript and now state the following: “Higher leg and trunk muscle activation during balance control in copers versus people with chronic ankle instability and healthy female athletes”

Abstract:  

- Electromyography, they make use of acronyms without putting the word it refers to.

Response: The description of the acronym has now been provided we now state “Surface electromyography (EMG)….”

- It is said that the Cumberland ankle instability tool was used as part of the method, but it was only used for group discrimination.

Response: We have now removed the Cumberland ankle instability tool from the abstract.

- In form it is well written, it deals with all the points of the work.

Response: We thank the reviewer for their supportive comment

 Introduction

- At the beginning of the introduction reference is made to an injury rate for 2010 and only in the United States, but this is not indicated in the text and gives the impression that it is a global rate.

Response: We now state the following with a reference change.

“Ankle sprains are one of the most prevalent musculoskeletal injuries with athletes in the United States reporting ankle sprain accounting for 15% of total injuries”

- Reference 7 is not well arranged in the text, it is not properly arranged and it is not clear what it refers to. It talks about postural control in unstable situations rather than in unstable situations. It is placed before talking about specific muscular activations of both tibialis anterior and gluteus maximus, to a generalization regarding the neuromuscular receptors of the ankle, which would explain the changes in the muscular activation previously described.

Response: We have now updated the referencing in this section.

- The comprehension of the last paragraph is difficult due to the absence of semicolons.

Response: We have now revised the paragraph so that it is more comprehensible

-Some more references are needed to support what is said "Individuals with CAI usually show changes in the activation of the muscles..............." and to support the 3 hypotheses that are described prior to the hypothesis that is the object of the study.

Response: We have now added some referencing in this section.

”Individuals with CAI usually show changes in the activation of the muscles around the ankle and sometimes the proximal muscles, and also during changes in functional tasks, including balance. ”

- Regarding the structure of the introduction, it should be improved to speak from the most general to the most particular, the first paragraph is in the correct situation of the introduction, but the second, in part should be almost at the end of the introduction before launching the hypothesis and study question.

Response: We have now placed the second paragraph before the study question and hypothesis

  1. Materials and Methods

2.1 Participants

-The inclusion criteria are well founded, with the scientific literature.

Response: Thank you for the supportive comment

- In the exclusion criteria, criterion number 3 is missing, it goes from criterion 2 to criterion 4.

Response: We have now revised the number in our exclusion criteria

-Exclusion criterion number 4, is it checked in any way?

Response:  The information was collected on the datasheet and any consumption of alcohol or drugs resulted in the exclusion of the participant.

-The specification of in which leg the measurement of each group is carried out should not be in this section of the research, but in the following one.

Response: We have now transferred the information to the following section.

2.2 Experimental procedure

- It would be good to indicate the place where the study was carried out, laboratory, it is supposed, but it is not specified.

Response: Thank you! We have included the place where the study was conducted. We now state “This study was carried out at the Laboratory of Health and Sports Medicine Department, University of Tehran, Tehran, Iran.”

- The clothing and footwear used by the participants, both in the warm-up and in the performance of the tests, are not specified.

Response: We now state the following

“All participants wore sports clothes and shoes during warm-up, and short tops and shorts during measurements.”

- It is not specified how is the familiarization of the participants with the BBS test, which may give more time to adapt to the test between participants.

Response: We now state the following

“After the warmup, participants were familiarized with the balance tests at two levels of difficulty (level 3: unstable and level 12: stable) on the Biodex Balance System (BBS) system. All participants were given a practice session using the same testing protocol, as a familiarization with the balance platform.”

- The part of the text that refers to the MVC of the EMG should be described in section 2.4 Electromyography (EMG) measurement. 

Response: The information has been transferred to the EMG section

-Figure 1 as above should be located in section 2.4 Electromyography (EMG) measurement.

Response: The figure has been placed in the requested section

2.3 Postural stability

- The specification of the footwear should be in the previous section 2.2 Experimental procedure.

Response: Specification of the footwear has been added and we now state

“All participants wore sports clothes and shoes during warm-up, and short tops and shorts during measurements.”

- What happens if the postural stability criteria are not met? Is the test repeated or is the test null? Not specified.

Response:  We now added the following statement

“If the postural stability was not met during the test, the results from the related trial were nullified and the participant repeated the trial once more.”

- Does the angle of foot placement on the platform not matter?

Response: We have added the following information

“The foot was positioned in the centre of the platform, following the guidelines specified by BBS. Foot position was recorded and replicated for each person for each trial.”

2.4 Electromyography (EMG) measurement

- This section is well explained.

Response: Thank you

- For a better reproduction of the study and better assessment, the exact location for each of the muscles could be known and not only for the electrodes used, but also for the electrodes used in the study. 

Response: We now state “…muscles following SENIAM guidelines as illustrated in figure 2”

- The SENIAM guide is referenced, but only to refer to the electrodes used.

Response: We have added the following statement

“The electrodes (self- adhesive Ag/AgCl bipolar, SKINTACT ECG Electrodes, Austria) were placed on the TA, MG, RF, BF, ES, and RA muscles following SENIAM guidelines”

- Add the section on calculation of the CVM and find out whether the CVM and the test were performed on the same day, taking advantage of the same location of the CVM for the subsequent performance of the test.

Response: The following information has been added

“Finally, the MVCs from the TA, MG, RF, BF, ES, and RA muscles were obtained using the method described previously by Konrad on the same day.”

2.5 Statistical analysis

-This section is correctly described

Response: Thank you!

  1. Results

3.1 EMG activity during unstable condition

- Unstable situations are mentioned in the title of the section, but data are shown in Table 2 for both unstable situations (Level 3) and stable situations (level 12).

Response: We would like to thank the reviewer for their comment. In order to prevent exceeding number of tables, the data for the unstable and stable conditions have been amalgamated into one table (i.e., table 2) and it appears at the end of the first paragraph that has been mentioned.

- Only 3 muscles are mentioned, Medial gastrocnemius (MG), Rectus abdominis (RA) and Rectus femoris (RF); the remaining muscles are omitted and it is not specified whether or not there are differences in these muscles between groups.

Response: We have now added the following statement

“No significant difference was found between groups for TA, BF, and ES muscle activity during unstable conditions (p > 0.05)”

- It does not follow a line in the presentation of the data, the MG compares it between coper and healthy with CAI, the RA compares coper versus CAI and RF compares healthy with CAI.

Response: In Table 1 the mean and SD EMG data are represented for unstable (level 3) and stable (level 12) conditions. For example, if you consider the MG muscle for the coper group and compare it with healthy group, it can be noted that the coper group demonstrated higher (133%) muscle activity compared to the healthy group which is in line with the data presented in the table.

3.2 EMG activity during stable condition

- The part of table 2 referring to stable situations (Level12) should be indicated in this section.

Response: See previous comments

- The font format used in this section is not as it should be, it is in italic format as if it were an image caption.

Response: The italic font has been corrected.

- Only 3 muscles are mentioned, Tibialis anterior (TA), Rectus abdominis (RA) and Rectus femoris (RF); the remaining muscles are omitted and it is not specified whether or not there are differences in these muscles between groups.

Response: We have now added the following statement

“No additional differences were noted for other muscles during the stable condition (p >0.05).”

- does not follow a line in the presentation of the data, the TA compares it between coper and healthy with CAI, the RA compares coper versus CAI and RF compares healthy with CAI.

Response: See previous comment

3.3 EMG activity between stable and unstable conditions

- The values of MG are missing in table 3, since the values of stable conditions (Level 12) are not found in any of the tables or results.

Response: The MG values have been added to table 3

- no differences are specified in healthy subjects for rectus femoris (RF) no significant differences are found.

Response: Since there were no significant differences, details regarding this data are not reported either in the table or text.

- no differences are specified for erector spinae (ES) for coper subjects.

Response: Since there were no significant differences, details regarding this data are not reported either in the table or text.

3.4 Balance error scores during unstable condition 

- Only comparison data between CAI and healthy subjects are presented; some reference to copers is missing.

Response: We have added to following statement for clarity

“No further differences were observed between the CAI, coper and healthy groups (p>0.05)”

- A table of results can be added, to facilitate the understanding of the data.

Response: Table 4 contains mean scores for stable and unstable conditions for all groups and it has been moved up to this section.

3.5 Balance error scores during stable conditions

- Only comparison data between copers and healthy subjects are presented, some reference to CAI is missing.

Response: Response: We have added to following statement for clarity

“No further difference was observed between the CAI, coper and healthy groups (p>0.05)”

- A results table can be added for these data to improve their visualization.

Response: Table 4 contains mean scores for stable and unstable conditions for all groups

3.6 Balance error scores between stable and unstable conditions

-This point is well defined.

Response: Thank you!

  1. Discussion

- It talks about the decrease in EMG for CAI in single leg stance, without specifying whether in stable or unstable conditions or both.

Response: Further clarification has been provided and we now state the following

“Regarding changes in surface stability, the unstable surfaces compared to stable surfaces, were found to induce greater muscle activation in all groups.”

-It refers to EMG of joints (ankle, knee, hip) rather than muscle groups of those body areas.

Response: Further clarification has been added to define the muscle groups that are being referred to.

- In the fourth paragraph reference is made to other results that support the findings of this study, but no references are specified to support that statement.

Response: A study by Jaber et al. (2018) has been added to support our claim

  1. Conclusions

- This point of the paper is well written except for specifying what type of single leg support it refers to, whether it is stable or unstable or both, besides the medial gastrocnemius no data is shown regarding single leg support in a stable situation in any part of the paper.

Response: Further clarification regarding the conditions and muscles have been provided in this section.

  1. Bibliography

This section is correctly expressed.

General analysis 

Congratulations to the authors for the study, the subject is very interesting and affects a large population. 

It is suggested that a revision of the material and methods be made, being more exhaustive and concrete in the explanations of the use of electromyography and better detailing the procedures for performing the tests.

In addition, in the results section it is proposed to improve the presentation of the data, not leaving comparisons between groups without making them and following a better line to improve the understanding of the data and make the article easier to understand.

It would also be good to introduce a section on limitations and lines of future research or to add it at the end of the conclusion. 

Response: We have now added some suggested lines of future research.

“One limitation of this study was the lack of use of both sexes. Inclusion of male coper participants and comparing them to female coper population may suggest better as to how neuromuscular activity among different lower extremity muscles compare between sexes. “

Reviewer 2 Report

Journal Sports (ISSN 2075-4663)

Manuscript ID sports-1785886

Title: Higher leg and trunk muscle activation during balance control in copers versus people with chronic ankle instability

In general, the current study was to examine different muscle activity associated with trunk and lower limb of female athletes during balance tests, mainly to report an adapted chronic ankle instability (CAI) the coper group’s closed to that of healthy group, compared to CAI group whose EMG activity decreased might contribute to impaired balance. Although the findings of the study may provide some insights to the coper related researches, major information was lack to provide and some minor comments are given as the following:

Major comments

1. Line 68~74, the three hypotheses were not strongly supported by the literature reviews due to lack of reporting current studies found in Watabe et al. (2022& 2021 seen below), and most references reviewed in this study seems out of date.    

·         Watabe, T., Takabayashi, T., Tokunaga, Y., & Kubo, M. (2022). Copers adopt an altered dynamic postural control compared to individuals with chronic ankle instability and controls in unanticipated single-leg landing. Gait & Posture, 92, 378-382. doi:https://doi.org/10.1016/j.gaitpost.2021.12.014

·         T. Watabe et al.Individuals with chronic ankle instability exhibit altered ankle kinematics and neuromuscular control compared to copers during inversion single-leg landing Phys. Ther. Sport. (2021)

·         T. Watabe et al.Copers adopt an altered movement pattern compared to individuals with chronic ankle instability and control groups in unexpected single-leg landing and cutting task J. Electromyogr. Kinesiol. (2021)

 2.      Line 267~275: The conclusion section is not consistently and clearly to indicate the coper group whose increased activity level of the muscles might result in more trunk and ankle stability, which is reported at the end of the current abstract.

Minor comments

Line 48~81, the paragraph is too long to be read and should be broken according to the ideas, especially for the coper based reviews.

Line 101: Table 1, some biased statistical results due to unequal sample size among the testing groups (copers, n=10; CAI, n=13; and healthy athletes, n=11) should be fixed or mentioned for a proper statistical method in the 2.5 Statistical analysis session.  

Line 139. “Each EMG value was calculated using the rate of maximum voluntary contraction (%MVC)” The rate of MVC is not clear.

Line 170~172. The entire paragraph is incorrectly italicized.

Line 175~177. The sentence is awkward to read although readers can guess the meaning of this written statement.

Line 224~225: The unclear statement can be fixed to show the CAI group’s EMG activity. “In contrast, one study reported greater MG EMG activity [of ?] compared to the copers and healthy group.”

Line 225~229: The contents related to the Kwon’s result are not clearly reported for the author his finding, such that the CAI group showed greater of the FL and MG than the coper and healthy groups. [In the Kwon study (2018), it reveals that the CAI group demonstrated greater NMAs of the FL and MG than the healthy and coper groups in the eye closed (EC).]

Line 214~216, “…for proximal muscle groups (MG) and distal (RF and RA) to the ankle ….” is clear for the anatomical position statement, but for the Line 268~270, “… of muscles both proximal (RF and RA) and superior (MG) to the ankle during single-leg stance,…” may not be clear for its consistency.

Author Response

In general, the current study was to examine different muscle activity associated with trunk and lower limb of female athletes during balance tests, mainly to report an adapted chronic ankle instability (CAI) the coper group’s closed to that of healthy group, compared to CAI group whose EMG activity decreased might contribute to impaired balance. Although the findings of the study may provide some insights to the coper related researches, major information was lack to provide and some minor comments are given as the following:

Major comments

  1. Line 68~74, the three hypotheses were not strongly supported by the literature reviews due to lack of reporting current studies found in Watabe et al. (2022& 2021 seen below), and most references reviewed in this study seems out of date.   
  •          Watabe, T., Takabayashi, T., Tokunaga, Y., & Kubo, M. (2022). Copers adopt an altered dynamic postural control compared to individuals with chronic ankle instability and controls in unanticipated single-leg landing. Gait & Posture, 92, 378-382. doi:https://doi.org/10.1016/j.gaitpost.2021.12.014
  • T. Watabe et al.Individuals with chronic ankle instability exhibit altered ankle kinematics and neuromuscular control compared to copers during inversion single-leg landingPhys. Ther. Sport. (2021)
  • T. Watabe et al.Copers adopt an altered movement pattern compared to individuals with chronic ankle instability and control groups in unexpected single-leg landing and cutting taskJ. Electromyogr. Kinesiol. (2021)

Response: We have now added the suggested references along with their results in our introduction. We have added the following statement

“The CAI group has shown altered activity of the muscles supporting the ankle due to nerve and muscle damage. Recent studies reported that CAI individuals significantly showed less peroneus longus activity and shorter peroneus longus latency compared to the Coper group in dynamic tasks. In the study of Kwon et al., they showed that the muscle activity in the CAI group is different compared to the healthy and Coper groups during single-leg balance with eyes closed.”

  1. Line 267~275: The conclusion section is not consistently and clearly to indicate the coper group whose increased activity level of the muscles might result in more trunk and ankle stability, which is reported at the end of the current abstract.

Response: Thank you for the suggestion. Muscle names have been provided in the last statement of the conclusion statement to be in line with our abstract conclusion. We now state:

“Increased EMG activity of the MG, TA, and RA in copers might be contribute to compensatory mechanisms, which resulted in greater trunk and ankle stability.”

Minor comments

Line 48~81, the paragraph is too long to be read and should be broken according to the ideas, especially for the coper based reviews.

Response: Revisions have been made to the paragraph that breaks down the information for readers.

Line 101: Table 1, some biased statistical results due to unequal sample size among the testing groups (copers, n=10; CAI, n=13; and healthy athletes, n=11) should be fixed or mentioned for a proper statistical method in the 2.5 Statistical analysis session. 

Response: Thank you for bringing this to our attention. We have conducted a Brown-Frosythe F-test due to unequal sample sizes between groups for our demographic data. The results of the first table have been updated as well as the information in the statistical analyses section. We now state the following:

“The Brown-Frosythe F-test was used to compare the difference in height, weight and body mass index (BMI) between three groups due to unequal sample sizes between groups.”

Line 139. “Each EMG value was calculated using the rate of maximum voluntary contraction (%MVC)” The rate of MVC is not clear.

Response: Thank you for the comment. We have revised the statement and now state: “EMG values were normalized to their respective peak root-mean-square value during an MVC (EMG value/MVC %).”

Line 170~172. The entire paragraph is incorrectly italicized.

Response: The italic format has been corrected

Line 175~177. The sentence is awkward to read although readers can guess the meaning of this written statement.

Response: The sentence has now been revised and we now state:

“EMG muscle activity of the TA in the CAI (↑76%) and healthy (↑64%) groups; BF in the CAI (↑39%) group; RF in the CAI (↑46%) and coper (↑47%) groups; ES in the CAI (↑54%) and healthy (↑11%) groups; and RA in the CAI (↑6%) and coper (↑12%) groups had significantly higher EMG activity from level 12 (stable) compared to level 3 (un-stable) BBS (Table 3).”

Line 224~225: The unclear statement can be fixed to show the CAI group’s EMG activity. “In contrast, one study reported greater MG EMG activity [of ?] compared to the copers and healthy group.”

Response: The statement has been revised and we now state the following: “In contrast, one study reported greater MG EMG activity in CAI group compared to the copers and healthy group.”

Line 225~229: The contents related to the Kwon’s result are not clearly reported for the author his finding, such that the CAI group showed greater of the FL and MG than the coper and healthy groups. [In the Kwon study (2018), it reveals that the CAI group demonstrated greater NMAs of the FL and MG than the healthy and coper groups in the eye closed (EC).]

The CAI group demonstrated greater NMAs of the PL and MG than the healthy and coper groups in the EC.

Response: The statement has been revised and now we state the following:

“In their study, Kwon et al. reported during a single-leg balance test with eyes closed, the CAI group had higher fibularis longus and MG activity than the healthy and coper groups.”

Line 214~216, “…for proximal muscle groups (MG) and distal (RF and RA) to the ankle ….” is clear for the anatomical position statement, but for the Line 268~270, “… of muscles both proximal (RF and RA) and superior (MG) to the ankle during single-leg stance,…” may not be clear for its consistency.

Response: Thank you for the constructive comment. We have revised the sentence accordingly and we now state: “The results of this study demonstrated that the EMG activity of CAI group was lower than the other two groups during single-leg stance. Individuals with CAI showed a decrease in EMG activity of muscles both distal (RF and RA) and proximal (MG) to the ankle during single-leg stance in the unstable condition, and it is difficult to conjecture with certainty, which strategy in these individuals is more likely to contribute to maintaining the balance with single-leg stance. Also, people with CAI had balance deficits in the medial-lateral direction. Reduced muscle activity in individuals with CAI could be one of the reasons for the balance deficits in these individuals. Increased EMG activity of the MG, TA, and RA in copers might be contribute to compensatory mechanisms, which resulted in greater trunk and ankle stability.”

Reviewer 3 Report

This study aims at grasping characteristics of muscle activations during balance tasks among three groups of people; normal, coper and those with CAI. The manuscript is clearly written and easy to follow in general. There are a couple of clarifications needed to increase the readability of this manuscript. 

Major points:

1) The assessment of performance 

BBS error is used for the purpose of evaluating the balance performance of the subjects. Since not all the readers are familiar with this variable, a short description of this variable would be helpful. 

Comparison among the groups is not easy to see. Are CAI and COPER bad in balance? 

2) Conflicting result

Decreased muscle activities in CAI group is somehow counterintuitive for me and therefore discussion of Kwon's study is very interesting. However, the effect of vision may not cut it unless the interaction of vision and ankle condition is elaborated. 

3) Highlights: neuromuscular deficits

Since all the subjects successfully complete the task, what is observed in this study is the result of an adaptation to given conditions. I am not sure how to conclude 'deficit' in neuromuscular control. 

Minor points:

1) page 2, line 84, ' ...were randomly allocated to three groups:...'

Can't be done. 

2) Page 3:

Figure 3 is not really informative and may be removed. 

Author Response

1) The assessment of performance 

BBS error is used for the purpose of evaluating the balance performance of the subjects. Since not all the readers are familiar with this variable, a short description of this variable would be helpful. Comparison among the groups is not easy to see. Are CAI and COPER bad in balance? 

Response: Thank you for the suggestion. We have now added the following statement: “BBS measures the degree of tilt about each axis during dynamic action in closed-chain conditions, and provides stability indices, and calculates anterior/posterior index (API), medial–lateral stability index (MLI) and the overall balance index (OBI).”

2) Conflicting result

Decreased muscle activities in CAI group is somehow counterintuitive for me and therefore discussion of Kwon's study is very interesting. However, the effect of vision may not cut it unless the interaction of vision and ankle condition is elaborated. 

Response: We have added and revised the following statement: “The lack of visual input has been known to disrupt balance, thus requiring more muscular contribution to achieve stability compared to having visual feedback as individuals would rely more on proprioception feedback mechanism for balance which leads to altered muscle activation.”

3) Highlights: neuromuscular deficits

Since all the subjects successfully complete the task, what is observed in this study is the result of an adaptation to given conditions. I am not sure how to conclude 'deficit' in neuromuscular control. 

Response: Thank you for the comment. Failure of the task does not necessarily represent neuromuscular deficiency rather the amount, or lack of (i.e., decreased), muscle activation can also signify a deficit in the neuromuscular activation even though the task at hand has been completed.

Minor points:

2) Page 3:

Figure 3 is not really informative and may be removed.

Response: Thank you for your suggestion. The figure has been removed.

Round 2

Reviewer 2 Report

Thanks for the revision.